# The rise of grasslands is linked to atmospheric $CO_2$ decline in the late Palaeogene

Luis Palazzesi [1,2✉], Oriane Hidalgo[2,3], Viviana D. Barreda[1], Félix Forest[2,6] & Sebastian Höhna[4,5,6✉]

Grasslands are predicted to experience a major biodiversity change by the year 2100. A better understanding of how grasslands have responded to past environmental changes will help predict the outcome of current and future environmental changes. Here, we explore the relationship between past atmospheric $CO_2$ and temperature fluctuations and the shifts in diversification rate of Poaceae (grasses) and Asteraceae (daisies), two exceptionally species-rich grassland families (~11,000 and ~23,000 species, respectively). To this end, we develop a Bayesian approach that simultaneously estimates diversification rates through time from time-calibrated phylogenies and correlations between environmental variables and diversification rates. Additionally, we present a statistical approach that incorporates the information of the distribution of missing species in the phylogeny. We find strong evidence supporting a simultaneous increase in diversification rates for grasses and daisies after the most significant reduction of atmospheric $CO_2$ in the Cenozoic (~34 Mya). The fluctuations of paleo-temperatures, however, appear not to have had a significant relationship with the diversification of these grassland families. Overall, our results shed new light on our understanding of the origin of grasslands in the context of past environmental changes.

[1] Museo Argentino de Ciencias Naturales & Consejo Nacional de Investigaciones Científicas y Técnicas (CONICET), Buenos Aires C1405DJR, Argentina. [2] Jodrell Laboratory, Royal Botanic Gardens, Kew, Richmond, Surrey TW9 3DS, UK. [3] Institut Botánic de Barcelona (IBB, CSIC-Ajuntament de Barcelona), Catalonia, Spain. [4] GeoBio-Center, Ludwig-Maximilians-Universität München, Richard-Wagner-Str. 10, 80333 Munich, Germany. [5] Department of Earth and Environmental Sciences, Paleontology & Geobiology, Ludwig-Maximilians-Universität München, Richard-Wagner-Str. 10, 80333 Munich, Germany. [6]These authors contributed equally: Félix Forest, Sebastian Höhna ✉email: lpalazzesi@macn.gov.ar; hoehna@lmu.de

The grassland biome (steppes, savannas, and prairies) covers vast areas of the Earth's surface and today accounts for as much as one-third of the net primary production on land[1,2]. Although grasses (Poaceae) comprise the bulk of the biomass and plant population in grasslands, other plant families—in particular the daisies (Asteraceae)—are usually as much as (or even more) diverse than grasses (Supplementary Fig. 1). The evolution of grasslands marked the emergence of a new landscape and provided the substrate for the adaptive radiation of other life forms that coevolved along with this biome, including grazing mammals[3] such as horses, wombats, and capybaras.

The age of a given biome is often estimated by detecting when particular representative taxonomic groups first appear in the fossil record. For example, the early evolution of the grassland biome—and open-habitat biomes in general—has been estimated from the fossil record of grass phytoliths (plant silica)[4] or from the record of fossil pollen of daisies, grasses, and amaranths[5,6]. Phylogenetic trees based on DNA sequence data calibrated with fossils provide a powerful new perspective on the history of biomes[7]. This approach has been used to estimate the timing of tropical-rainforest evolution based on phylogenetic trees of plant groups that are characteristic of this biome (e.g., Malpighiales[8], Arecaceae[9], and the legume genus *Inga*[10]). Nevertheless, phylogenetic approaches have barely been used to study the evolutionary history of grassy biomes; most previous studies of grassland evolution have focused on the origins of $C_4$ grasslands[11]. Here we estimate when grasslands first expanded using phylogenetic trees of its two primary plant families, Poaceae and Asteraceae. We assembled a large calibrated phylogenetic tree for daisies and used the largest tree yet inferred for grasses[11] to explore temporal shifts in rates of lineage diversification, and to test correlations between diversification-rate shifts and past climatic fluctuations.

A major limitation when analyzing hyper-diverse groups—in our case Asteraceae with ~23,000 species and Poaceae with ~11,000 species—is the inevitable sparse species sampling (Figs. 1 and 2). Although existing approaches for inferring rates of lineage diversification (speciation and extinction) can accommodate incomplete species sampling[12,13], the distribution of missing species on the tree in these approaches is modeled in a simplistic and somewhat unrealistic manner. Previous works have shown that biased species sampling has a strong impact on diversification-rate estimates[14–16]. Here, we develop a Bayesian approach for detecting diversification-rate shifts that incorporates a more realistic (non-uniform) model of species sampling and implemented it in the open-source software RevBayes[17]. Our model builds on the episodic birth–death process, where speciation and extinction rates are constant within an interval but may shift instantly to new rates at a rate-shift episode[18–21]. This model assumes that diversification rates are homogeneous (equal for all lineages at the same time) and does not allow for lineage-specific shifts in diversification rates. Furthermore, we test for a correlation between diversification rate and two environmental variables —atmospheric $CO_2$ concentration and average global paleo-temperature— using one existing[22–27] and three here developed environmentally-dependent diversification models. We use an empirically informed and biologically realistic model to accommodate missing species that assigns unsampled species to their corresponding clades using taxonomic information.

## Results and discussions

Our analyses demonstrate that the most dramatic increase in diversification rates in both Asteraceae and Poaceae (calibration scenario #1, see "Methods") occurred from the late Oligocene (~28 Mya) to the early Miocene (~20 Mya) (Fig. 3 and

Supplementary Fig. 6). This diversification rate shift is robust to several model assumptions. We recovered the same diversification rate shifts regardless of the assumed number of time intervals (Supplementary Fig. 7). Both autocorrelated diversification rate prior models qualitatively agree on the overall pattern of diversification rates (Gaussian Markov random field (GMRF) or Horseshoe Markov random field (HSRMF), Supplementary Figs. 6 and 7). Only the uncorrelated diversification rate prior model differed in the inferred pattern (UCLN, Supplementary Figs. 6 and 7). However, the autocorrelated diversification rate prior models were significantly favored according to our Bayes factor analyses (GMRF for the daisy phylogenetic tree and HSMRF for the grass phylogenetic tree, Supplementary Fig. 8). Recently, Louca and Pennell[28] showed that phylogenies of extant taxa are consistent with infinitely many diversification rate models and therefore diversification rates are not identifiable if arbitrarily complex diversification rate functions are allowed. Our diversification models, on the other hand, are identifiable because of the piecewise-constant (episodic) diversification rates model[29]. Furthermore, model comparison is robust when well-formulated alternative hypotheses are used[30], as is the case for the comparison between different environmentally dependent diversification models[31].

The diversification rate patterns were strongly influenced by the assumed incomplete taxon sampling (Supplementary Fig. 9). In our simulation study we show that incorrectly assuming uniform taxon sampling and thus disregarding taxonomic information about the distribution of missing species strongly biases diversification rates (Supplementary Figs. 22 and 23). Conversely, our empirical taxon sampling informed by a more accurate distribution of missing species has good power to detect the correct time-varying diversification rates and low false-positive rate when diversification rates are in reality constant (Supplementary Fig. 23). Thus, we recommend to include as much information as possible regarding the distribution of missing species.

The respective diversification rates of Asteraceae and Poaceae (calibration scenario #1, see "Methods") peak between 20 Mya and 15 Mya, and subsequently decreases for a brief period of time before increasing again from the late Miocene (~10 Mya, Fig. 3 and Supplementary Figs. 4–6). Our second analysis using the Poaceae phylogeny calibrated with a Cretaceous phytolith (calibration scenario #2) detects an earlier peak for Poaceae at about 35–30 Mya (Supplementary Fig. 6). The phylogenetic placement of this fossil phytolith has been debated[32], however here we show the results of the two alternative hypotheses (calibration #1 and calibration #2) proposed by Christin et al.[32] rather than selecting one over the other; other works on Poaceae have also adopted a similar approach (e.g. Hackel et al.[33]). Previous works on Asteraceae and Poaceae identified clades with increased diversification rates using calibrated molecular phylogenies; for example, Mandel et al.[34] detected the highest acceleration rates in the Vernonioid clade (Cichorioideae) and within the Heliantheae alliance of the Asteraceae family, both at the early Miocene (~23 Mya) using MEDUSA[35]. They also detected other lineages with relatively high rates at the late Eocene (~40 Mya). Previously, Panero & Crozier[36] also found the most important shifts in diversification along these two lineages (i.e. Vernonioid clade and Heliantheae alliance) using BAMM[37]. On the other hand, Spriggs et al[11] found twelve shifts using their calibrations scenarios #1 and #2 on Poaceae. They detected clades with the highest diversification rates during the Neogene (23–2.4 Mya) using turboMEDUSA[38]. Overall, the timing of the diversification-rate shifts identified by our model broadly agrees with the shifts recognized for the clades with the highest rates according to the previously published data. However, our results clearly indicate that the diversification rate shift occurred while all major subfamilies diversified simultaneously

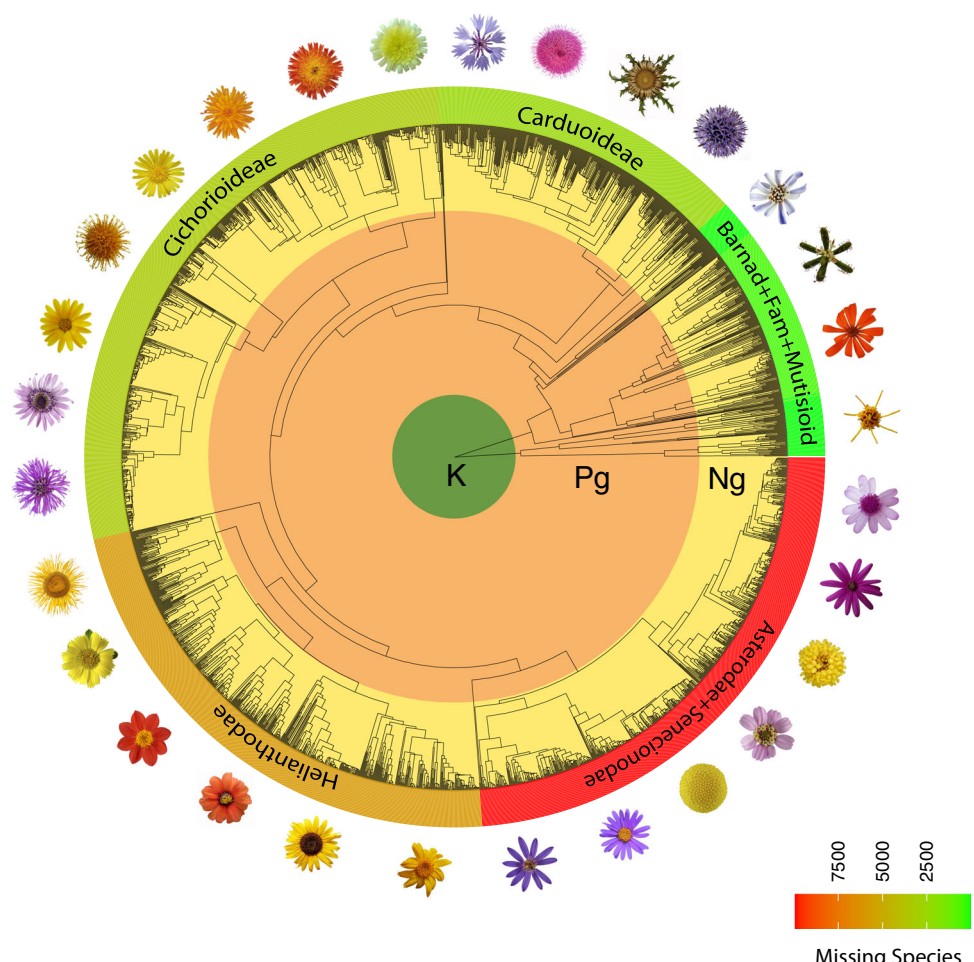

**Fig. 1 Phylogenetic tree scaled to geological time of Asteraceae with 2723 sampled tips.** Asteraceae is one of the most species-rich families of flowering plants with more than 23,000 species. The number of non-sampled (missing) species increases enormously towards the more derived and species-rich lineages. For this reason, the sampling among clades is severely biased. Note that these rich and derived lineages evolved during the late Paleogene or early Neogene. K Cretaceous, Pg Paleogene, Ng Neogene.

(Supplementary Fig. 4 and 5), thus indicate rather a global (i.e., tree-wide) pattern than a lineage-specific effect.

Interestingly, estimations of low diversification rates prior to ~35Mya are consistent with the scarcity of fossil forms assigned to both daisies (Supplementary Table 1) and grasses[4,39]. Similarly, high diversification rates during or after the Oligocene are in line with the high diversity of fossil remains assigned to these groups[4,40]. The Cenozoic 'temporal hotspot' of grassland diversification (~30 Mya to ~15 Mya) based on daisies and grasses (calibration #1 and #2) phylogenetic trees—coincides with one of the most fundamental changes in global climate in the geologic record; a marked decline of atmospheric $CO_2$ occurred during the Oligocene (~34 Mya), reaching modern levels by the latest Oligocene[41,42]. This scenario marks the onset of a cooler and more modern world (Coolhouse state), identified by the earliest Cenozoic glaciations in Antarctica, and the consequent drop in global paleo-temperatures[43].

In line with the reconstructed climatic scenario, our analyses of correlation between diversification rates and $CO_2$ or paleo-temperature show very interesting results (Fig. 3 and Supplementary Fig. 10 and 11). Diversification rates inferred from both the daisy and grasses phylogenies support correlation to $CO_2$ over paleo-temperature (Supplementary Fig. 11). Surprisingly, the best fitting environmentally-dependent diversification model for the daisy phylogeny was the uncorrelated lognormal (UCLN)

variation model and for the grasses phylogeny the *fixed* rate model without additional variation. The support of the uncorrelated model over the two autocorrelated models (GMRF and HSRMF), although the autocorrelated models were favored when using time-varying diversification rates without environmental variables (Supplementary Fig. 8), could stem from the use of vague prior distribution which allows for more rate variation in autocorrelated models[21]. However, regardless of the specific environmentally dependent diversification model, we inferred a negative correlation between diversification rates and environmental $CO_2$ (Supplementary Fig. 10). The resulting Bayes factors for a negative correlation were decisive with values of 37,501 for the *fixed*, UC and GMRF models and 49 for the HSMRF model (Fig. 3). We also see the same agreement between the four environmentally-dependent diversification models in our simulation study (Supplementary Fig. 20 and 21). Thus, if there is a clear signal of correlation between the environmental variable and diversification rates, then our analyses appear robust to modeling of the additional component of time-varying diversification rates. This agreement can also be seen when all four environmentally-dependent diversification models show the same estimated diversification rates (Supplementary Figs. 14 and 15). When the signal is less clear, as for the paleo-temperature analyses, the four models disagree and range from significant positive to significant negative correlation (Supplementary Fig. 10) and the estimated

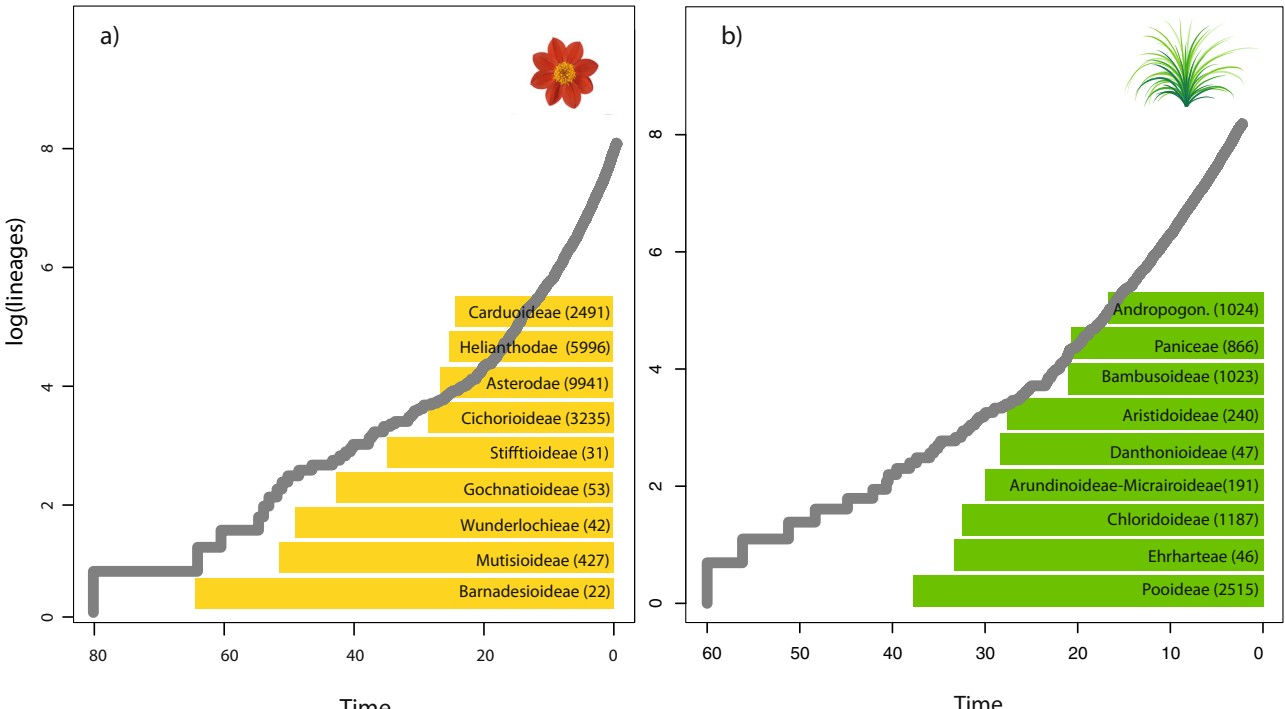

**Fig. 2 Lineage-Through Time (LTT) plots of daisies and grasses.** Solid gray lines represent LTT curves derived from time-calibrated phylogenetic trees of **a** Asteraceae and **b** Poaceae (calibration scenario #1). Colored boxes depict the name and number of non-sampled (missing) species per clade that we integrated in our empirical taxon sampling. The shape of LTT curves have demonstrated to be a convenient summary metric for diversification diagnostics, particularly when diversification deviates from the expectation of constant rates[81,82]. However, the distribution of missing species might not be uniform— as it is the case of these angiosperm families—and can severely impact on diversification-rate estimates. Our work shows that the most important increase in diversification rate for both Asteraceae and Poaceae is completely unnoticeable using the LTT analysis, even when calibrated phylogenetic trees include a large number of species.

diversification rates of the environmentally-dependent diversification models also differ (Supplementary Figs. 14 and 15). Finally, our results of correlation between environmental $CO_2$ and diversification rates are also robust to the chosen epoch size (Supplementary Fig. 12).

The negative correlation between diversification rates of these selected grassland families and atmospheric $CO_2$ might not be surprising; atmospheric $CO_2$—the main source of carbon for photosynthesis—serves as a fundamental substrate for plant growth. The available experimental evidence shows that low atmospheric $CO_2$ limits plant performance[44], although responses vary significantly between species. At a landscape scale, carbon limitation and water stress due to lower atmospheric $CO_2$ concentrations ('ecophysiological drought'), rather than water stress due to lower precipitation ('climatic drought'), cause changes in vegetation structure[45]. During the Last Glacial Maximum (LGM; ~21,000 years ago), for example, atmospheric $CO_2$ was at its lowest concentration in the history of land plants (~180–200 ppm)[46]. Models have predicted that the direct physiological impact of the of low $CO_2$ concentrations during the LGM drove the expansion of grasslands and dry shrublands at the expense of forest[47] (Supplementary Fig. 2). Other modeling experiments indicate that low atmospheric $CO_2$, in combination with increased aridity and decreased temperatures, causes new xeric biomes to develop[46].

Although our primary hypothesis is that a $CO_2$-depleted atmosphere played a role in the geographic expansion and diversification of grassland families from Oligocene times (~34 Mya), other environmental and biological variables could have also been involved. In particular, the decreasing temperatures, increasing aridity, and increasing seasonality of temperature and/or precipitation of the late Cenozoic have been traditionally linked to the

early radiation of grasslands[48,49]. The role of cooling in the emergence of open-habitat grasses has been debated as the adaptation to low temperatures became prominent in the more derived groups of grasses[4,50]. Grazing mammals also have been important components in the evolution of grasslands; grazers and grassland ecosystems probably coevolved over millions of years[51]. Grazing increased species diversity according to experimental studies, as grazers prevent dominant plant species from monopolizing resources. Without grazing, tall, vegetatively reproducing plant species increase in cover and shade out short and sexually reproducing species[52]. Grazing also affects the flux of nutrients by accelerating the conversion of plant nutrients from forms that are unavailable for plant uptake to forms that can be readily used. Overall, grazing mammals have an important role in the diversity of present-day natural grasslands and we assume they might have done so during their early radiation. However, the explosive radiation of true hypsodonts may have negatively impacted grasslands' distribution and diversity (see below). Sorting out the relative importance of all these environmental and biological competing forces from the hypothesized $CO_2$-induced shift remains challenging.

We detected a short decrease in diversification rates for daisy and grass plant groups during the mid-Miocene, about 13–10 Mya (Figure 3a). The causal mechanism underlying remains to be elucidated. However, we suspect that the dramatic radiation of hypsodont grazers—such as horses—and other mixed feeder grazers may have had an impact on grasslands[3,53]. Since the late Miocene (~10 Mya), however, the more recent expansion of $C_4$ grass lineages[11] may have contributed to the increased diversification rates in these groups. Plants using the $C_4$ photosynthetic pathway have anatomical and biochemical adaptations for concentrating $CO_2$ within leaf cells prior to photosynthesis, which may lead to a selective advantage over $C_3$

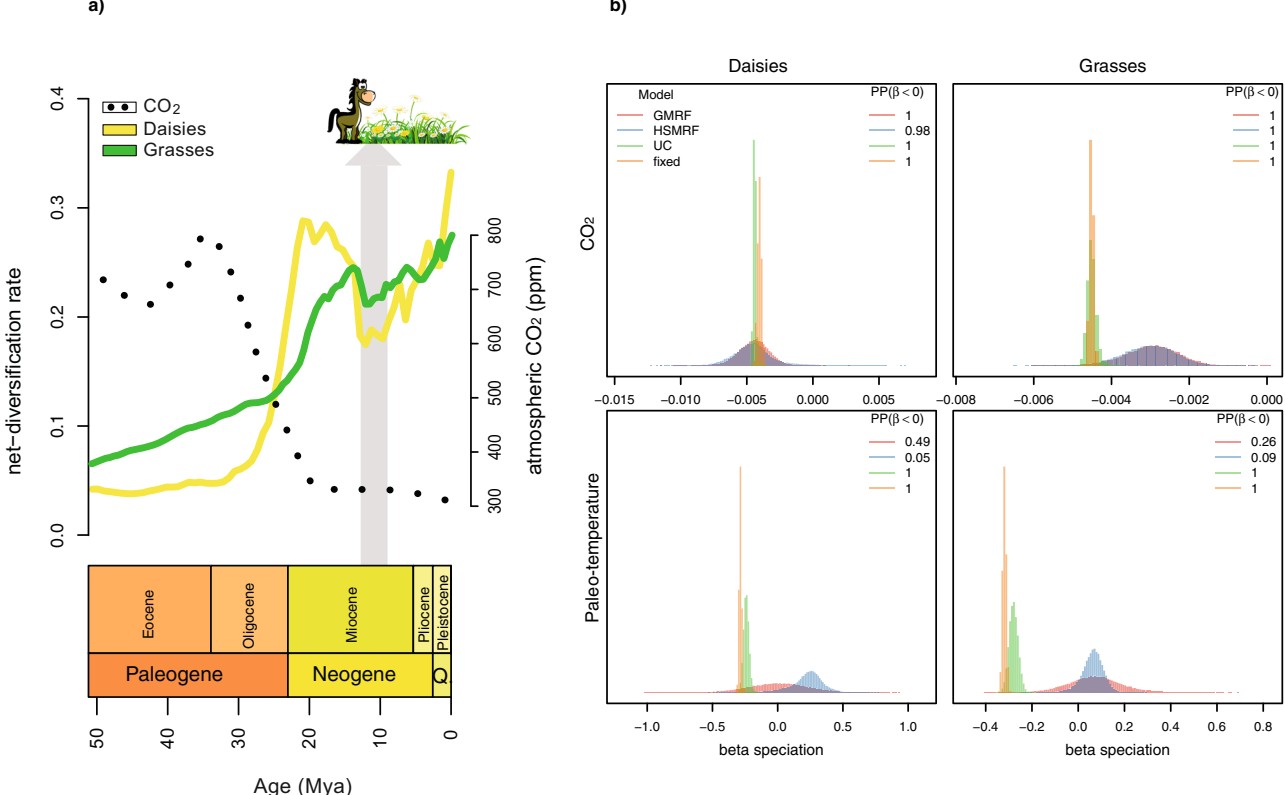

**Fig. 3 Estimating diversification rates and correlation to $CO_2$ and paleo-temperature. a** Diversification rates through time of daisies (yellow) and grasses (green)—calibration scenario #1—for the last 50 Mya. Dotted line represents atmospheric $CO_2$ fluctuations (22); note that the Oligocene steep decrease mirrors the onset of the increase in diversification of daisies and grasses. Gray arrow indicates a period (13–10 Mya) of lower diversification rates probably linked either with a brief increase in $CO_2$ (not represented in the dotted smoothed curve) or the explosive radiation of hypsodont grazers (e.g., horses) and other mixed feeder grazers[3] who may have had a tremendous impact on grasslands through their effects on plant populations and community composition. **b** The correlation factor ($\beta$) between diversification rates and $CO_2$ concentrations for daisies and grasses is significantly negative (posterior probability of 1.0 for all models except the HSRMF, which has a posterior probability of 0.98; Bayes factors of 37,501 and 49 respectively). The support for a negative correlation between paleo-temperature and diversification rates is ambiguous; the UC and *fixed* models show significant support (posterior probability of 1.0, Bayes factor of 37,501) while the autocorrelated models show no support (posterior probability between 0.05 and 0.095, Bayes factors supporting a *positive* correlation of 1.04 and 17.86 for the daisy dataset and 2.85 and 9.75 for the grasses dataset for the GMRF and HSRMF models respectively). Note that the correlation factor $\beta$ does not follow the same scale as common correlation coefficients (which are between −1 and 1) but instead represents the factor by which to transform the environmental variable into diversification rates (see "Methods").

plants under conditions of low atmospheric $CO_2$. Although the evolutionary origin of $C_4$ photosynthesis in grasses most likely occurred early in the Cenozoic[32], their expansion and ecological dominance may have taken place during the last 10 Mya, by the late Miocene in warmer and fire-prone landscapes of the world[54]. Likewise, the evolution of hyper-diverse Asteraceae lineages (e.g., *Senecio*)[55] have also contributed to the increasing rates of diversification since the last 10 Mya. Our evidence also supports the notion that the ongoing rise of atmospheric $CO_2$ will likely alter vegetation distributions through differential effects on $C_3$ and $C_4$ plant types. In fact, modeling future distributions predicts the near-complete eradication of $C_4$ species across the globe for the next 50 years[56]; this implies that about half of the species in the grass family will be extinct. In summary, our study reveals episodic shifts in diversification rates of grasses and daisies which are correlated with changes in atmospheric $CO_2$ (Fig. 3a); these insights are made possible by the development of our Bayesian phylogenetic approach which combines the episodic birth–death process[18–21] with environmentally-dependent diversification rates[22–24,26] and empirical taxon sampling[15,16,57]. Our environmentally dependent and episodic birth–death diversification model provides an approach for exploring the evolution of hyper-diverse groups of plants and animals in the context of historical environmental changes.

## Methods

**Taxonomic representativeness in grasslands**. To quantify the taxonomic representativeness of vascular-plant families found in open-habitat landscapes (Supplementary Fig. 1), we selected seven distantly distributed eco-regions dominated by grasslands from the World Wide Fund for Nature[58]. Using the coordinate boundaries of each of the selected eco-regions, we extracted the vascular plant taxa (=Tracheophyta) using the R 'rgbif:Interface to the Global 'Biodiversity' Information Facility API' package[59] with the option 'hasGeospatialIssue=FALSE', that includes only records without spatial issues (e.g., invalid coordinates, country coordinate mismatch). Plant families were sorted according to the number of species, removing duplicated species.

**Palaeobotanical analysis**. Asteraceae and Poaceae have a fairly similar fossil record; their oldest findings are known from the Late Cretaceous—which mainly comprise microscopic remains (that is, phytoliths[60] or pollen grains[61])—whereas the first indisputable macroscopic Asteraceae and Poaceae fossils are first known from the Eocene[62,63], with a substantial increase of diversity since the Oligocene/Miocene. While the fossil record of Poaceae has been fully revised[4,64,65], the fossil record of Asteraceae has not been as carefully reviewed. We compiled published pollen and macroscopic fossil data for Asteraceae including all fossil species assigned to Asteraceae (Supplementary Table 1). The earliest record of the Asteroideae (the clade that includes the most common open-habitat daisy tribes) occurs since the Late Oligocene of New Zealand but in very low frequencies. Fossils refer to this subfamily increased in abundance and diversity during the Miocene and Pliocene. Pollen referred to *Artemisia*, in particular, did not become abundant until the Middle-Late Miocene with several reports from central Europe, Asia and North America. Pre-Miocene findings need further verification. Overall, the Late Oligocene and in particular the Miocene witnessed the major step in the diversification of Asteraceae; ca. 80% of the fossil species recorded have been assigned to this time interval.

**Divergence-time estimation.** To construct the Asteraceae supertree (2723 tips), we first inferred a backbone chronogram using 14 plastid DNA regions from 54 species, including representatives of all 13 subfamilies, with an additional four species of Calyceraceae used as outgroup taxa (Supplementary Table 2). Sequences were compiled from GenBank and each region was aligned separately using MAFFT[66] with the options maxiterate 1000 and localpair. Two fossil constraints were applied: (i) a macrofossil (capitulum) and associated pollen (*Raiguenrayun cura* + *Mutisiapollis telleriae*) from the Eocene (45.6 Mya) to calibrate the non-Barnadesioideae Asteraceae clade[63] and; (ii) the fossil-pollen species *Tubulifloridites lilliei* type A from the late Cretaceous (72.1 Mya)[61] to calibrate the crown Asteraceae (considering *T. lilliei* as a stem group, see Huang et al.[67] for further discussion). Divergence-time estimates and phylogenetic relationships were inferred using `RevBayes`[17]. For the aligned molecular sequences we assume a general-time reversible substitution model with gamma-distributed rate variation among sites (GTR+Γ), an UCLN prior on substitution-rate variation across branches (UCLN relaxed clock), and a birth–death prior model on the distribution on node ages/tree topologies. A densely sampled phylogeny is crucial to identify shifts in diversification rates. Therefore, we constructed a supertree by inserting eleven individual sub-trees —representing all subfamilies of the Asteraceae except those less diverse or monotypic clades (that is, Gymnarrhenoideae, Corymbioideae, Hecastocleidoideae, Pertyoideae)—into the calibrated backbone chronogram. This method follows a previous study that constructed a supertree of grasses using the same approach[11]. Each of the eleven clades of Asteraceae was built using their own set of markers and the same phylogenetic approach as the one used to infer the backbone tree (Supplementary Table 2). Sequence data for each of the eleven trees and their respective outgroup taxa were collected from Genbank using the NCBIminer tool[68]. The estimated ages of the nodes given by the backbone analysis were used to constrain the age of each of the eleven sub-trees (Supplementary Table 2). Divergence-time estimates and phylogenetic relationships for each of the eleven sub-clades were estimated using `RevBayes` as described above. The eleven trees were grafted onto the backbone tree using the function 'paste.tree' from the phytools R package[69]. We used ggtree R package[70] to plot the circle phylogenetic tree of Figure 1 and phytools[69] to include the concentric geological scale. The supertree of the grass family (3,595 taxa) was obtained from Spriggs et al.[11] (Supplementary Table 3). They inferred two chronograms using two different calibration scenarios, that is, a younger scenario (#1) calibrated using an Eocene megafossil[62] and an older scenario (#2) calibrated using Cretaceous phytoliths[60]. We ran our diversification analyses using these two chronograms.

**Inferring changes in diversification rate through time.** Our species-diversification model is based on the *reconstructed evolutionary process* described by Nee et al.[12] and more specifically on the episodic birth–death process[18–21]. We assume that each lineage gives birth to another species with rate $\lambda$ (cladogenetic speciation events) and dies with rate $\mu$ (extinction event; see Fig. 4). We model diversification rates (i.e., speciation and extinction rates) as constant within an interval but independent between intervals, where intervals are demarcated by instantaneous rate-shift events, and equal among contemporaneous lineages. We denote the vector of speciation rates $\Lambda = \{\lambda_1, \ldots, \lambda_k\}$ and extinction rates $\mathbf{M} = \{\mu_1, \ldots, \mu_k\}$ where $\lambda_i$ and $\mu_i$ are the (constant) speciation and extinction rates in interval $i$. Additionally, we use the taxon-sampling fraction at the present denoted by $\rho$[15,16]. Following the notation of May et al.[20], we construct a unique vector, $\mathbb{X}$, that contains all divergence times and rate-shift event times sorted in increasing order. It is convenient for notation to expand the vectors for all the other parameters so that they have the same number of elements $k = |\mathbb{X}|$. Let $\Psi$ denote an inferred tree relating $n$ species, comprising a tree topology, $\tau$, and the set of branching times, $\mathbb{T}$. We use the notation $S(2, t_1 = 0, T)$ to represent the survival of two lineages in the interval $[t_1, T]$, which is the condition we enforce on the reconstructed evolutionary process. Transforming Equation (A4) in May et al.[20] to our model yields the probability density of a reconstructed tree as:

$$f(\Psi | N(t_1 = 0) = 2, S(2, t_1 = 0, T))$$
$$= \frac{2^{n-1}}{n!}$$
$$\times \left(1 + \sum_{i=0}^{k} \left(\frac{\mu_i}{\mu_i - \lambda_i} \times e^{\sum_{j=0}^{i-1}(\mu_j - \lambda_j)(x_{j+1} - x_j)} \times \left(e^{(\mu_i - \lambda_i)(x_{i+1} - x_i)} - 1\right)\right) - \frac{\rho - 1}{\rho} \times e^{\sum_{i=0}^{k}(\mu_i - \lambda_i)(x_{i+1} - x_i)}\right)^{-2}$$
$$\times \left(e^{-\log(\rho)\sum_{j=0}^{k}(\mu_j - \lambda_j)(x_{j+1} - x_j)}\right)^2$$
$$\times \prod_{i \in \mathbb{T}} \left[\lambda_i \times \left(1 + \sum_{l=i}^{k} \left(\frac{\mu_l}{\mu_l - \lambda_l} \times e^{\sum_{j=0}^{l-1}(\mu_j - \lambda_j)(x_{j+1} - x_j)} \times \left(e^{(\mu_l - \lambda_l)(x_{l+1} - x_l)} - 1\right)\right)\right.\right.$$
$$\left.\left. - \frac{\rho - 1}{\rho} \times e^{\sum_{i=i}^{k}(\mu_i - \lambda_i)(x_{i+1} - x_i)}\right)^{-2} \times e^{-\log(\rho)\sum_{j=i}^{k}(\mu_j - \lambda_j)(x_{j+1} - x_j)}\right].$$
$$(1)$$

The first term, $\frac{2^{n-1}}{n!}$ corresponds to the combinatorial constant for the number of labeled histories[18], the second term corresponds to the condition of two initial lineage at the root of the phylogeny surviving until the present, and the third term corresponds to the product of all speciation events and the new lineages surviving until the present.

**Empirical taxon-sampling model.** Here we develop an *empirical* taxon-sampling model that uses taxonomic information on the membership of unsampled species to clades and speciation times of unsampled species, which is an extension to the work by Höhna et al[15,16] and similar to the approach used by Stadler and Bokma[57]. The main difference of our approach and the approach by Stadler and Bokma[57] is that their model uses a constant-rate birth–death process (compared to our episodic birth–death process). Additionally, Stadler and Bokma[57] derive the density of the missing species using a random probability $s$ of an edge being sampled, which differs from our approach where we integrate over the time of the missing speciation event. Nevertheless, at least for the constant-rate birth–death process, both approaches arrive at the same final likelihood function.

We include information on the missing speciation events by integrating over the known interval when these speciation events must have occurred (that is, between the stem age $t_c$ of the MRCA of the clade and the present, Figure 4). This integral of the probability density of a speciation event is exactly the same as one minus the cumulative distribution function of a speciation event[16],

$$F(t_c | N(t_1) = 1, t_1 \le t \le T) = 1 - \frac{1 - P(N(T) > 0 | N(t_c) = 1) \exp(r(t_c, T))}{1 - P(N(T) > 0 | N(t_1) = 1) \exp(r(t_1, T))}, \quad (2)$$

where $t_1$ is the age of the root. The probability of survival is given by:

$$P(N(T) > 0 | N(t_c) = 1)$$
$$= \left(1 + \sum_{i=c}^{k} \left(\frac{\mu_i}{\mu_i - \lambda_i} \times e^{\sum_{j=c}^{i-1}(\mu_j - \lambda_j)(x_{j+1} - x_j)} \times \left(e^{(\mu_i - \lambda_i)(x_{i+1} - x_i)} - 1\right)\right) - \frac{\rho - 1}{\rho} \times e^{\sum_{i=c}^{k}(\mu_i - \lambda_i)(x_{i+1} - x_i)}\right)^{-1}$$
$$(3)$$

where $k = |\mathbb{X}|$. Let us define $n$ as the number of sampled species, $m$ as the total number of species in the study group, $\mathbb{K}$ as the set of missing species per clade and $|\mathbb{K}|$ the number of clades with missing species. Additionally, we define $c_i$ as the time of most recent common ancestor of the $i^{th}$ clade. Then, the joint probability density of the sampled reconstructed tree and the empirically informed missing speciation times is

$$f(\Psi, \mathbb{K} | N(t_1 = 0) = 2, S(2, t_1 = 0, T)) = f(\Psi | N(t_1 = 0) = 2, S(2, t_1 = 0, T))$$
$$\times \frac{(m-1)!}{(n-1)!} \prod_{i=1}^{|\mathbb{K}|} \frac{1}{k_i!} \left(1 - F(t | N(c_i) = 1, c_i \le t \le T)\right)^{k_i}.$$
$$(4)$$

**Prior models on diversification rates.** Our model assumes that speciation and extinction rates are piecewise constant but can be different for different time intervals (Fig. 4). Thus, we divide time into equal-length intervals (e.g., $\Delta t = 1$). Following Magee et al.[21], we specify prior distributions on the log-transformed speciation rates ($\ln(\lambda_i)$) and extinction rates ($\ln(\mu_i)$) because the rates are only defined for positive numbers and our prior distributions are defined for all real numbers. We apply and compare three different prior models: (i) an UCLN prior distribution, (ii) a GMRF prior[21], and (iii) a HMRF prior[21]. The first prior distribution specifies temporally uncorrelated speciation and extinction rates, whereas the second and third prior distributions are autocorrelated prior models. The assumption of autocorrelated rates might make more sense biologically (an interval of high speciation rates is likely to be followed by another interval with high speciation) but also improves our ability to estimate parameters[21]. Nevertheless, our inclusion of both uncorrelated and autocorrelated prior distributions allows for testing whether an uncorrelated or autocorrelated model is preferred.

The prior distribution on the speciation rates $\lambda_i$ and extinction rates $\mu_i$ are set in exactly the same form in our models with their respective hyperprior parameters. Thus, for the sake of simplicity, we omit the prior distribution on the extinction rates here in the text. Our first prior distribution, the UCLN distributed prior, specifies the same prior probability for each speciation rate $\lambda_i$,

$$\ln(\lambda_i) \sim \text{Normal}(m, \sigma). \quad (5)$$

Thus, each speciation rate is independent and identically distributed.

Our second prior distribution, the GMRF prior, models rates in an autocorrelated form analogous to a discretized Brownian motion. That is, we assume that diversification rates $\lambda(t)$ and $\mu(t)$ are autocorrelated and the rates in the next time interval will be centered at the rates in the current time interval,

$$\ln(\lambda_i) \sim \text{Normal}(\ln(\lambda_{i-1}), \sigma_\lambda). \quad (6)$$

The standard deviation $\sigma$ regulates the amount of change between each time interval.

Our third prior distribution, the HSMRF prior, is very similar to the GMRF but additionally allows for the variance to change between time intervals,

$$\gamma_i \sim \text{halfCauchy}(0, 1) \quad (7)$$

$$\ln(\lambda_i) \sim \text{Normal}(\ln(\lambda_{i-1}), \sigma\gamma_i). \quad (8)$$

The HSMRF prior model is more adaptive than the GMRF; it allows for more extreme jumps between intervals while favoring/smoothing more constant-rate trajectories if there is no evidence for rate changes[21].

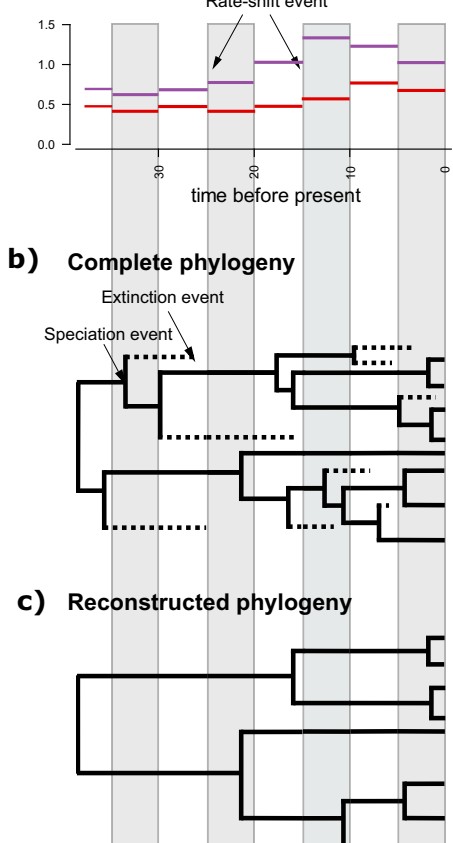

**a) Diversification rates through times**

Rate-shift event

time before present

**b) Complete phylogeny**

Extinction event

Speciation event

**c) Reconstructed phylogeny**

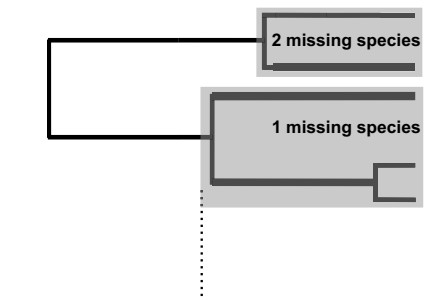

**d) Sampled phylogeny with missing species**

2 missing species

1 missing species

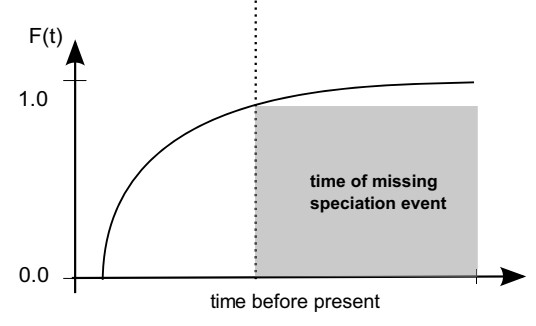

**e) Distribution function of missing speciation event**

$F(t)$

1.0

0.0

time of missing
speciation event

time before present

**Fig. 4 Cartoon of the birth–death process with rate-shift events and empirical taxon sampling. a** Depiction of the speciation (purple lines) and extinction (red lines) rates through time. Here we assume that speciation and extinction rates are episodically constant, that is, diversification rates shift instantly and only at the beginning of an episode. Each episode lasts five time units in this example. **b** A realization (complete phylogeny) of the birth–death process. Lineages that have no extant or sampled descendant are shown as dashed lines and surviving lineages are shown as solid lines. **c** Reconstructed phylogeny corresponding exactly to the one shown in B with the extinct lineages pruned away. Thus, plot **c** depicts the "observed" phylogeny from which the speciation times are retrieved. **d** Sampled phylogeny with gray boxes depicting named clades with known number of missing species. The phylogeny is the same as in c with fewer taxa. **e** Distribution function of the time of the missing speciation event. The missing speciation event could have occurred any time between the crown age of the named clade and the present time (gray box). The distribution function is integrated over and hence the uncertainty of the missing speciation event accounted for.

These three prior models of diversification rates provide the null models of our analyses as they do not assume any dependence to an environmental variable. We use these models first to estimate diversification rates through time before testing for a correlation of the speciation or extinction rate to an environmental variable (e.g., atmospheric $CO_2$ or paleo-temperature). Magee et al.[21] found that 100 epochs perform well for autocorrelated models. Since we do not know how many bins (i.e., epochs) should be used for the episodic birth–death process, we test various numbers of equal-sized epochs (4, 10, 20, 50, 100, and 200, Supplementary Figure 7). We show both the median posterior diversification rates (Supplementary Fig. 6) as well as select the best fitting model based on the number of epochs (Supplementary Fig. 8).

**Correlation between speciation and extinction rate to $CO_2$.** Previously, Condamine et al.[22] introduced an environmentally-dependent diversification model. In their model, diversification rates are correlated with an environmental variable[22–27]. For example, the speciation rate can be modeled as $\lambda(t) = \lambda_0 e^{\beta \times CO_2(t)}$ (see Box 1 in Condamine et al.[22]), which is equivalent to $\ln(\lambda(t)) = \ln(\lambda_0) + \beta \times CO_2(t)$. Since we are using the episodic birth–death process which has piecewise-constant diversification rates, we modify the original continuous-time environmentally-dependent diversification model to $\ln(\lambda_i) = \ln(\lambda_0) + \beta \times CO_{2,i}$, which is equivalent to and more conveniently written as $\ln(\lambda_i) = \ln(\lambda_{i-1}) + \beta \times \Delta CO_{2,i}$ where $\Delta CO_{2,i} = CO_{2,i} - CO_{2,i-1}$. Note that we only use the so-called exponential dependency and not the linear dependency[24] because the linear dependency can result in negative rates which are mathematically and biologically impossible[71].

We applied this original environmentally-dependent diversification model and three environmentally-dependent diversification models described in this work. The original environmentally-dependent diversification model of Condamine et al.[22] does not accommodate diversification-rate variation that is independent of the environmental variable. Instead, our three environmentally-dependent diversification models build on our diversification-rate prior models which allow for rate variation through time (see above). Thus, our environmentally-dependent diversification models will collapse to the episodic birth–death model if rates of diversification and atmospheric $CO_2$ are uncorrelated and hence inherently allows for diversification rate variation. The linkage of environmental variable and diversification rates without allowing for independent diversification rate variation might provide spurious results, as has been noticed for trait evolution[72] and state-dependent diversification rates[73]. We explore this potential of misattribution of diversification rate variation to the environmental variable in our model selection procedure and simulation study (see below).

As before, we omit the description of the extinction rates in the text for the sake of notational simplicity. Both speciation and extinction rates are model exactly in the same way with their corresponding set of hyperparameters (e.g., see the Supplementary Tables 4–7). Our first environmentally-dependent diversification model has a *fixed* linkage between the diversification rate variation and variation in the environmental variable;

$$\lambda_0 \sim \text{Uniform}(0, 100) \tag{9}$$

$$\ln(\lambda_i) = \ln(\lambda_{i-1}) + \beta_\lambda \times \Delta CO_2. \tag{10}$$

This model does not have a counterpart in the above diversification rate priors, but is included as a comparison to the work Condamine et al.[22].

Our second environmentally-dependent diversification model adds UCLN variation on top of the variation in the environmental variable;

$$\lambda_0 \sim \text{Uniform}(0, 100) \tag{11}$$

$$\ln(\hat{\lambda}_i) = \ln(\hat{\lambda}_{i-1}) + \beta_\lambda \times \Delta CO_2 \tag{12}$$

$$\epsilon_i \sim \text{Normal}(0, \sigma) \tag{13}$$

$$\ln(\lambda_i) = \ln(\hat{\lambda}_i) + \epsilon_i. \tag{14}$$

Thus, this model collapses to the above UCLN model if there is no correlation between the environmental variable and diversification rates ($\beta = 0$). Importantly, the difference in the variation of the diversification rates and environmental variable is independent in each epoch, contributed by the variable $\epsilon_i$. The environmental-dependent part of the diversification rates $\hat{\lambda}_i$ is equivalent to the *fixed* environmentally-dependent diversification model.

Our third environmentally dependent diversification model adds correlated lognormal variation on top of the *fixed* environmentally-dependent diversification mode;

$$\lambda_0 \sim \text{Uniform}(0, 100) \tag{15}$$

$$\ln(\lambda_i) \sim \text{Normal}(\ln(\lambda_{i-1}) + \beta_\lambda \times \Delta CO_2), \sigma). \tag{16}$$

This model an extension of the above GMRF model and collapses to it if there is no correlation between the environmental variable and diversification rates ($\beta = 0$). As the GMRF model is a discretized Brownian motion model, this environmentally-dependent extension can be considered as a Brownian motion with trend model, where the trend is predicted by the environmental variable. Instead of writing this model with a separate environmentally-dependent part $\hat{\lambda}_i$ and autocorrelated part $\epsilon_i$, we directly use the combined environmentally-dependent and independent rate variation as the mean for the next time interval. Nevertheless, we want to emphasize this equivalence to bridge the connection to the UCLN model above.

Finally, our fourth environmentally-dependent diversification model extends the above HSRMF to allow for diversification rates predicted by the environmental variable;

$$\lambda_0 \sim \text{Uniform}(0, 100) \tag{17}$$

$$\gamma_i \sim \text{halfCauchy}(0, 1) \tag{18}$$

$$\ln(\lambda_i) \sim \text{Normal}(\ln(\lambda_{i-1}) + \beta_\lambda \times \Delta CO_2), \sigma\gamma_i). \tag{19}$$

This model follows the same extension as the environmentally-dependent GMRF model with local adaptability of the rate variation through the parameter $\gamma_i$, as before for the HSMRF.

In all our four models, we denote the correlation factor by $\beta$. If $\beta > 0$ then there is a positive correlation between the speciation rate and $CO_2$, that is, if the $CO_2$ increases then the speciation rate will also increases. By contrast, if $\beta < 0$ then there is a negative correlation between the speciation rate and $CO_2$, that is, if the $CO_2$ concentration increases then the speciation rate will decrease. Finally, if $\beta = 0$ then there is no correlation and our environmentally-dependent diversification model collapses to corresponding episodic birth–death model.

All four models have the same parameter for the initial speciation rate $\lambda_0$ with a uniform prior distribution between 0 and 100. The models are constructed in increasing complexity and all three models can collapse either to the *fixed* environmentally-dependent diversification model or to their environmentally independent episodic birth–death process.

**Environmental data**. In our analyses we tested for correlation between two environmental factors: $CO_2$ and temperature. The concentration of atmospheric $CO_2$ throughout the Cenozoic were compiled by Beerling & Royer[41] using terrestrial and marine proxies. An updated dataset was provided by Dr. Dana Royer. Paleo-temperature fluctuations come from Zachos et al.[74]. Raw data were extracted from ftp://ftp.ncdc.noaa.gov/pub/data/paleo/.

Analogous to our tests about the number of epochs for the diversification rate analyses, we computed the arithmetic mean for the environmental variable for 1-, 2- and 5-million year intervals. We both estimated the correlation between the environmental variable and diversification rates for each interval size and performed model selection using Bayes factors.

**Model selection**. We performed three sets of empirical diversification rate analyses for each dataset. We estimated the diversification rates over time using three different models, we estimated the environmentally-dependent diversification rates using four different models, and we applied two different taxon-sampling schemes. For the first two sets of analyses we performed standard model selection in a Bayesian framework using Bayes factors[75]. Thus, we computed the marginal likelihood for each model using stepping-stone sampling[76] as implemented in RevBayes[77]. We run 128 stepping stones with each stone comprising of its own

MCMC run with 2000 iteration and on average 1374 moves per iteration (i.e., the runs being equivalent to standard single-move-per-iteration software with 2,748,000 iterations).

We tested the support for the environmental correlation using Bayes factors computed from the posterior odds. Our prior probability for the correlation factor $\beta$ was symmetric and centered at zero, that is, we specified exactly a probability of 0.5 that $\beta < 0$ and $\beta > 0$. Thus, the prior probability ratio of $\frac{P(\beta<0)}{P(\beta>0)} = 1.0$ Then, to compute the Bayes factor for in support of a negative correlation is simply the number of MCMC samples with $\beta < 0$ divided over the total number of MCMC samples.

We did not compute marginal likelihoods for the two different sampling schemes; the uniform taxon sampling and the empirical taxon sampling. Empirical taxon sampling uses additional data, the age ranges of the missing speciation events, and two analyses with different data cannot be compared using traditional model selection. Instead, we performed a simulation study to show the robustness of our parameter estimates under empirical taxon sampling and the resulting bias if wrongly uniform taxon sampling was assumed.

**Simulation study**. We performed two sets of simulations; focusing (a) on the environmentally-correlated diversification model, and (b) the incomplete taxon same scheme. First, we simulated phylogenies under the UCLN and GMRF environmentally-correlated diversification model using the R package TESS[78,79]. We set the diversification rate variation to $\sigma = \{0, 0.02, 0.04\}$ and correlation factor to $\beta = \{0, -0.005, -0.01\}$. Thus, our simulations included the constant-rate birth–death process (when $\sigma = 0$ and $\beta = 0$), time-varying but environmentally independent diversification rates (when $\sigma > 0$ and $\beta = 0$), the *fixed* environmentally-dependent diversification model (when $\sigma = 0$ and $\beta \neq 0$), and the time-varying and environmentally-dependent diversification model (when $\sigma > 0$ and $\beta \neq 0$). For each setting, we simulated ten diversification rate trajectories (Fig. S16 and S17) and trees (Fig. S18 and S19). We analyzed each simulated tree under the same four environmentally dependent diversification model as in our empirical analysis (see above).

Second, we simulated phylogenetic trees under empirical taxon sampling to validate the correctness of our model derivation. Unfortunately, simulation of empirical taxon sampling is not straight forward. We circumvented the problem by randomly adding the missing species to the daisy phylogenetic tree, then drawing new divergence times under (a) a constant-rate birth–death process, and (b) a time-varying episodic birth–death process with rates taken from the empirical estimates. Then, we pruned the additional species to mimic empirical taxon sampling. The simulations under the constant-rate birth–death process provide information about falsely inferring diversification rate variation (false positives) and the simulations under the time-varying episodic birth–death process provide information about the power to correctly inferring diversification rate variation (power analysis). We simulated 100 trees under each setting and analyzed each tree using the GMRF prior model with both empirical and uniform taxon sampling. The MCMC inference settings were identical to the empirical analyses.

**Reporting summary**. Further information on research design is available in the Nature Research Reporting Summary linked to this article.

## Data availability
The authors declare that all data supporting the findings of our study are available within the article and its supplementary information files or upon request to the authors. Time-calibrated phylogenies from grasses and daisies used in the present study are deposited at https://doi.org/10.5061/dryad.74b5d and Supplementary Data 1, respectively.

## Code availability
Both models, the episodic birth–death process and the environmentally-dependent diversification model, are implemented in the Bayesian phylogenetics software RevBayes version 1.1.1[17]. Moreover, the implementation is not restricted to the models we introduce here because RevBayes is built on the principle of probabilistic graphical models[80]. The graphical model approach provides full flexibility to extend or modify the current analyses to other models and assumption, for example, testing for correlation to multiple environmental variables. RevBayes is open-source and freely available from https://revbayes.github.io/. The analysis from this paper are described in detail in several tutorials available at https://revbayes.github.io/tutorials/, specifically the tutorials https://revbayes.github.io/tutorials/divrate/ebd.html, https://revbayes.github.io/tutorials/divrate/env.html and https://revbayes.github.io/tutorials/divrate/sampling.html.

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

## Acknowledgements

We thank Brian Moore, Will Freyman, and Dana Royer for their comments on earlier versions of this manuscript. We also thank three anonymous reviewers who have greatly improved our work. LP received a Marie Curie International Incoming Fellowships from the European Union for project GRASSLANDS (Proposal 329652 & 912652 | FP7-PEOPLE-2012-IIF). Additional financial support was provided by ANPCyT and CONICET from the Argentinian Government. S.H. was supported by the Deutsche Forschungsgemeinschaft (DFG) Emmy Noether-Program HO 6201/1-1. This work was also supported by a BAYLAT-Anschubfinanzierung grant awarded to L.P. and S.H.

## Author contributions

L.P conceived the study, with input from S.H., F.F., O.H and V.D.B. S.H. and L.P. conceived and designed the experiments, performed the experiments, analyzed the data. L.P. and S.H. wrote the paper, with contributions from F.F., V.D.B and O.H. All authors approved the final manuscript.

## Competing interests

The authors declare no competing interests.
