## [Peer Review File · Nature Communications]

Reviewers' Comments:

Reviewer #1:

Remarks to the Author:

I was asked to review some of the methods in this manuscript, so my comments are confined to the diversification rate estimation and empirical taxon sampling.

The novel component of the diversification rate model is that speciation and extinction rates can shift due to unspecified factors. This presumably reduces incorrect inferences of associations between the focal variable and diversification. There is no description of how effective this is in practice (see below), but my guess is that it provides a significant improvement to a vexing problem. If so, it would be a valuable general advance that could apply to many other diversification models.

The novel way of dealing with incomplete taxon sampling is to integrate over the possible speciation times within the relevant clade. This seems like a clever approach, though again we're not shown how well it works in practice. Previous, less-Bayesian work has accomplished something similar by generating many possible trees with the missing tips attached, and then conducting inference across them. That has the disadvantage of weakening the diversification signal being sought. I just wonder if the integrative Bayesian approach in this manuscript might have the opposite problem: if missing taxa are placed in ways that are consistent with the model parameters being estimated, does that inflate confidence in the inferred diversification shifts? Perhaps this query merely shows that I don't think like a true Bayesian, but please explain (and show testing results) because I am surely not the only one to wonder.

Overall, I think these two methodological advances sound useful not only for the present study, but also as a basis for future work on related topics. However, there is one major omission which is not acceptable: there is no simulation (or other) testing of the new methods. How do we know that they work well? And that they work better than previous methods? Under what conditions? Even if a model is mathematically correct, it may not have suitable power and robustness when applied to empirical data, which is guaranteed not to follow the assumptions of the model. We need a thorough report of model performance (strengths and weaknesses) in order to know how much to trust the results in the present study, and for other researchers to consider if they wish to apply these new methods to their own data. This testing will be a lot of work to conduct and summarize, though some has presumably already been done (e.g., page 9, "We explored the approach of Condamine et al...").

Smaller issues:

Variables in the equation on pages 8-9 are not defined (b_i , d_i , ρ , n [only much later]). Please also provide a brief intuitive statement about what the key factors represent. Similarly for the math on pages 10-11 (t_c , t_1 , intuition).

Page 9: "The assumption of auto-correlated rates makes sense biologically..." Please explain why. Because speciation and extinction rates are governed by organismal traits and environments that vary smoothly and without directional bias over time?

Page 9: "...and also improves our ability to estimate diversification parameters." Improves in what sense? Inference is hopeless if this is not the true process? Inference is biased if other assumptions are made?

Page 10: "...age of the MRCA..." I think this actually means the stem group, based on $N(t_c) = 1$?

I cannot find anywhere an indication of how to obtain the code to run the new methods. This is not at all a "smaller" problem, but it was hopefully an accidental omission and easily remedied. The code (and sufficient documentation to use it) is essential to meet modern standards of reproducibility, and because one of the innovations of the work is a new method whose value is largely in its availability to other researchers.

Reviewer #2:

Remarks to the Author:

Palazzesi and colleagues present a phylogenetic study of two large plant clades (daisies and grasses), which are dominant in open-vegetation habitats. They build the largest Asteraceae super-tree to date and use a published phylogeny of Poaceae to investigate diversification rates through time. The authors develop a novel statistical method to assess temporal variation in speciation and extinction rates and test whether rate changes correlate with Cenozoic changes in atmospheric CO₂. They find that changes in speciation rates in both clades negatively correlate with changes in CO₂ (i.e. as CO₂ decreased, speciation rates increased).

The birth-death model developed in this paper models rate variation as a function of time (allowing rates to change between fixed bins in an auto-correlated fashion), and as a function of changes in CO₂ concentration (estimating correlation parameters between Δ CO₂ and speciation and extinction rates). This implementation clearly improves previous models correlating diversification rates with a time-variable curve in that the null model (i.e. when correlation parameters equal zero) is a more realistic birth-death with rate shifts, rather than a constant rate process.

While I think the manuscript is well written and the results are interesting, I also have some concerns about its design and methods. In general, I think the study should include more explicit hypothesis testing.

The assumption of auto-correlated speciation and extinction rates is justified by the fact that "it makes sense biologically" [p. 9]. While I agree that auto-correlated rates are a plausible hypothesis, the assumption seems to be in contrast with drastic rate changes linked to e.g. mass extinctions or adaptive radiations. The existence of autocorrelation should be something assessed by hypothesis testing. Is an auto-correlated model better than the alternative birth-death model with independent rates? For instance, one could implement a "white noise" version of the episodic birth-death model, with normal priors on the log-transformed rates and hyper-priors to adapt the prior distribution to the data. This issue is analogous to auto-correlation in molecular clock models. Clock rates across branches were assumed to be auto-correlated for years, and are now mostly considered uncorrelated based on empirical evidence.

The authors test here the effect of CO₂ on diversification without testing or discussing alternative explanations. For instance, diversity dependence and other biotic interactions could have contributed or even triggered the diversification of these groups. The author mention the effects of grassland expansion on grazer diversity, but I think reciprocal effects should be mentioned as well. Did the expansion of large herbivores (e.g. elephants) contribute to grass diversification by maintaining open habitats clear of forest or even contributing to their expansion? Temperature changes could also be a variable worth testing.

Alternative hypotheses (auto-correlation, independent-rates, CO₂, etc.) could be formally tested through Bayes factors. This would also help understanding by how much the CO₂ correlation improves the fit compared to a simpler episodic birth-death model.

The authors transform the environmental variable (CO₂ concentration) by computing the ratio between two sequential measurements and claim this approach "has the clear advantage that our computation is insensitive to the magnitude of the environmental variable". While I agree this is a good approach for CO₂ concentration, the suitability of this approach is still dependent on the variable: for instance using ratios for temperature would not make sense and would be scale dependent (e.g. Fahrenheit vs Celsius degrees). Furthermore, using the Δ -CO₂ in the analysis rather the CO₂ values, changes the interpretation of the correlation: it is not low CO₂ levels that explain increased speciation rates, rather it is the decrease of CO₂ that promotes diversification. What is the predicted speciation rate if CO₂ is low and constant?

Minor points

P. 8, first paragraph: I agree on discarding the second calibration scenario on the basis that the placement of microfossils is uncertain. However, the fact that the results of the second calibration scenario don't match the CO₂ dependence hypothesis (last sentence of the paragraph) should not be an argument for discarding the analysis.

I suggest adding a figure to show speciation and extinction rates through time as estimated under the model that includes both time-variable rates and CO₂ dependence. This will help understanding the magnitude of the effect of CO₂ on speciation rates.

What is the effect of bin size used in the episodic birth-death model on the inferred speciation and extinction rates and on the correlation factors?

Figure 2 shows a cartooned example of how the auto-correlated birth-death model works showing rate shifts between some of the time bins, whereas in other bins the rate is constant. This gives the impression that the model used here actually tests whether there is rate variation or not across bins, whereas –as far as I understand– rates are assumed to be independent across time bins.

Figure 3. "Daisies" is missing in the figure. Are the correlation coefficients (beta) between diversification-rates and CO₂ concentration relative to speciation (beta_lambda)?

The authors implemented a birth-death model to incorporate non-random incomplete taxon sampling, which has quite strong effects on the estimated rates compared to the standard assumptions. The implementation is described as "similar to the approach used by Stadler and Bokma", however it is not clear whether the method is in fact the one developed by Stadler and Bokma or if it differs from it.

It would be interesting to see in an extended figure the delta-CO₂ curve that was used to test for correlations between diversification and CO₂, i.e. the ratios between CO₂ at adjacent bins.

Adding equation numbers, page numbers, line numbers would help future revisions.

Reviewer #3:

Remarks to the Author:

This study presents a Bayesian phylogenetic approach that uses time-calibrated phylogenies to simultaneously estimate diversification rates through time and correlations between an environmental variable and lineage diversification. This method is then used to examine the influence of past CO₂ concentrations on the diversification of Poaceae and Asteraceae, two highly diverse grassland clades. They find accelerating diversification in both clades from ~28-15Ma, a short period of lower diversification rates and then further increases beginning ~10Ma.

This is a well-written paper and I like the overall approach that compares the diversification of Asteraceae and Poaceae to one another. The described method is powerful and will be useful for a wide variety of biological questions. I also appreciate the nuanced way that this paper discusses diversification—the authors never state that the drop in CO₂ concentrations drives diversification, only that they are correlated.

This analysis differs from previous studies of diversification in Poaceae in that it looks at shifts in diversification through time rather than among lineages. I would have liked to see a supplementary figure with diversification rates plotted on each phylogeny (Poaceae and Asteraceae)—maybe just simplified trees that have the major clades? It would then be possible to compare the timing of the shifts observed more directly to previous results. I also think that there should be some discussion about the simplifications that this model makes—specifically that all members of the clade are diversifying at the same rate at a given time. Both Asteraceae and Poaceae are diverse, cosmopolitan clades that contain a lot of physiological variation—a single diversification rate is probably summing over a lot of variation among lineages.

I appreciate that the authors have chosen not to focus on C4 photosynthesis, instead taking a wider view of grassland clades more generally; however, not including at least some discussion of how C4 photosynthesis is related to the observed patterns seems strange to me. C4 photosynthesis is generally believed to be an essential trait that has contributed to the global dominance of Poaceae and is highly related to plant performance under low atmospheric CO₂.

I had a few more specific comments as well:

"We selected the chronogram calibrated using the megafossil (calibrated scenario #1) because the phylogenetic placement of phytoliths is somewhat controversial¹⁶. . ."

The better reference for the implications/controversy of the phytolith Poaceae dates is Christin et al. 2014 (# 43). The choice here to use the macro fossil dating scenario seems reasonable and matches the choices made for Asteraceae, but I think the justification for this choice should be described better. It's an important decision because the phytolith dating scenario pushes back the ages for most of the Poaceae tree and it could significantly affect the results that are presented. With the older dates would the increase in Poaceae diversification predate the drop in CO₂? It seems from Figure 3 that it might not, but I think it would strengthen the argument of the paper if the results were robust to these dates. It's true that the phylogenetic placement of phytoliths is controversial because there is still uncertainty about the extent of homoplasy/their phylogenetic distribution, but they do provide useful information in clade with very little macro fossil information and discounting them entirely makes me uncomfortable.

"Furthermore, the origin of C4 lineages within Poaceae under the scenario #2 occurred well before the drop of CO₂. . ."

The only C4 origin that is well before the drop of CO₂ under dating scenario #2 is in the Chloridoideae. All other C4 origins occur near or after the CO₂ drop.

Figure 3. The yellow line (Asteraceae) is not labeled in the figure legend.

"The available experimental evidence from plants grown under low atmospheric CO₂ shows that plant performance deteriorates rapidly as CO₂ declines. . ."

More accurately low CO₂ limits performance of some taxa—responses vary significantly among species.

"The low CO₂ conditions probably imposed a selective pressure for life history traits; the main plant groups of the grassy biome—mostly annual or biannual herbs with short generation times. . . became extinct."

This section is confusing to me. Ecological models and experimental data suggest that high CO₂ favors woody plants over grasses (e.g. Kgope et al. 2009 'Growth responses of African savannah trees implicate atmospheric CO₂ as a driver of past and current changes in savanna tree cover; Bond & Midgley 2000 'A proposed CO₂ -controlled mechanism of woody plant invasion in grasslands and savannas; Bond & Midgley 2011 'Carbon dioxide and the uneasy interactions of trees and savannah grasses'), but this is separate from arguments related to generation time—Poaceae and Asteraceae species have many physiological adaptations to living in open environments. It makes sense, however, that short generation times might have enabled rapid evolution which could have facilitated their adaptation into new environments. Maybe this section would be clearer if 'open systems' or some alternative was used rather than 'grassy biomes.' It's hard to talk about grasses evolving to occupy biomes that are already characterized by their presence.

It would be useful if the ways that this method differs from previous methods (refs 22-24) were more explicitly stated. As it is, it's somewhat hard to tell.

Here, we provide a point-by-point response to all reviewers' concerns:

Reviewer #1 (Remarks to the Author):

I was asked to review some of the methods in this manuscript, so my comments are confined to the diversification rate estimation and empirical taxon sampling.

The novel component of the diversification rate model is that speciation and extinction rates can shift due to unspecified factors. This presumably reduces incorrect inferences of associations between the focal variable and diversification. There is no description of how effective this is in practice (see below), but my guess is that it provides a significant improvement to a vexing problem. If so, it would be a valuable general advance that could apply to many other diversification models.

Response: We appreciate the comments of Reviewer #1. We additionally added simulations and extended our model to other diversification models (see below).

Query #1) The novel way of dealing with incomplete taxon sampling is to integrate over the possible speciation times within the relevant clade. This seems like a clever approach, though again we're not shown how well it works in practice. Previous, less-Bayesian work has accomplished something similar by generating many possible trees with the missing tips attached, and then conducting inference across them. That has the disadvantage of weakening the diversification signal being sought. I just wonder if the integrative Bayesian approach in this manuscript might have the opposite problem: if missing taxa are placed in ways that are consistent with the model parameters being estimated, does that inflate confidence in the inferred diversification shifts? Perhaps this query merely shows that I don't think like a true Bayesian, but please explain (and show testing results) because I am surely not the only one to wonder.

Response: The reviewer raises a good point that perhaps several people might wonder about the accuracy of our incomplete taxon sampling approach. The accuracy actually should be independent of the statistical paradigm, Bayesian vs Maximum Likelihood estimates (however, this could only be tested if there would be an equivalent implementation of our model in a ML framework). In our model, we do not use any priors for the missing species and they simply contribute to the likelihood function. This is similar to all incomplete taxon sampling approaches (uniform and diversified taxon sampling), which have been applied both in Maximum Likelihood and Bayesian inference (Höhna 2014, Höhna et al., 2011).

To evaluate the accuracy and potential bias of the incomplete taxon sampling, we now perform a simulation study where we simulated both under a constant-rate diversification

model, and a time-varying diversification rates model. The constant-rate model answers the question if diversification rates are biased towards time-varying rates due to the assignment of missing taxa. We observe that if the model assumptions are met, that is, if we use the same incomplete taxon sampling approach for the inference as used in the simulations, then the diversification rates are robust. Conversely, if we wrongly assume uniform taxon sampling but in reality the tree was constructed using empirical taxon sampling, then the diversification rate estimates are biased (see Supplementary Figure S22, also shown here). These results are in line with our previous work on incomplete taxon sampling (Höhna 2014, Höhna et al., 2011).

We agree with the reviewer that this simulation study strengthens the manuscript and are glad that the reviewer has pointed this out to us.

Query #2) Overall, I think these two methodological advances sound useful not only for the present study, but also as a basis for future work on related topics. However, there is one major omission which is not acceptable: there is no simulation (or other) testing of the new methods. How do we know that they work well? And that they work better than previous methods? Under what conditions? Even if a model is mathematically correct, it may not have suitable power and robustness when applied to empirical data, which is guaranteed not to follow the assumptions of the model. We need a thorough report of model performance (strengths and weaknesses) in order to know how much to trust the results in the present study, and for other researchers to consider if they wish to apply these new methods to their own data. This testing will be a lot of work to conduct and summarize, though some has presumably already been done (e.g., page 9, "We explored the approach of Condamine et al...").

Response: We really appreciate Reviewer #1 feedback. We have extended our study in two main ways: (1) we included additional environmentally-dependent diversification models (see also the comments below), and (2) performed a simulation study based around the empirical parameter estimates.

In our simulation study, we included simulation with no rate variation, no correlation to the environmental variable, and different strength of correlation (see also new description in the Methods). We noticed that all four diversification models perform equally well, although the HSRMF model has the lowest uncertainty (Supplementary Figure S19 and S20, also reproduced below).

This final set of simulations included two sets of 360 MCMC analyses (simulations under uncorrelated and autocorrelated rates), where each analysis took about 2 days to run. We agree that robust testing is necessary, and here we show that the analyses perform as expected. We hope that the reviewer can appreciate that we cannot test every single parameter combination. We are surprised that we could not reproduce the difference in estimated correlation coefficients (see main Figure 3b), although this could be due to our use of the CO₂ instead of temperature as the environmental variable.

Smaller issues:

Query #3) Variables in the equation on pages 8-9 are not defined (b_i , d_i , ρ , n [only much later]). Please also provide a brief intuitive statement about what the key factors represent. Similarly for the math on pages 10-11 (t_c , t_1 , intuition).

Response: We have replaced b_i by λ_i and d_i by μ_i to clean up our notation. Now there is a brief description and introduction of the variables at the beginning of section "Inferring Changes in Diversification Rate Through Time". We have added some brief explanation of the terms after the equations, and also refer the reader to previous papers about more details.

Query #4) Page 9: "The assumption of auto-correlated rates makes sense biologically..." Please explain why. Because speciation and extinction rates are governed by organismal traits and environments that vary smoothly and without directional bias over time?

Response: We had previously assumed that the auto-correlated model best represented the changes in diversification rates through time without providing further evidence. There are two main reasons why auto-correlated rates work better. First, biologically it could be more plausible that in two short, consecutive intervals the speciation and extinction rates are more likely to be similar than not (we would not expect the rates to be completely arbitrary in consecutive intervals). Second, rate estimation is also more robust when the rates of the following interval are centered on the previous interval (see Magee et al 2020). After Reviewer #1 comments, we conducted Bayes factors, modified the text accordingly and discussed our selection in the context of the tested examples. Please, see Figure S7.

Query #5) Page 9: "...and also improves our ability to estimate diversification parameters." Improves in what sense? Inference is hopeless if this is not the true process? Inference is biased if other assumptions are made?

Response: The improved precision has been shown in Magee et al (2020), which we cite now. Additionally, we added our model selection procedure using Bayes factors to test if the rates should be autocorrelated or uncorrelated (see above).

Query #6) Page 10: "...age of the MRCA..." I think this actually means the stem group, based on $N(t_c) = 1$?

Response: Yes, we thank the reviewer for this catch.

Query #7) I cannot find anywhere an indication of how to obtain the code to run the new methods. This is not at all a "smaller" problem, but it was hopefully an accidental omission and easily remedied. The code (and sufficient documentation to use it) is essential to meet modern standards of reproducibility, and because one of the innovations of the work is a new method whose value is largely in its availability to other researchers.

Response: We absolutely agree with the reviewer that all code should be provided and open source. We now included the link where our package can be found and downloaded. For the new models and analyses in this paper, we have created specific tutorials which we host and maintain on our RevBayes website:

- Estimating time varying diversification rates using the GMRF and HSMRF models: <https://revbayes.github.io/tutorials/divrate/ebd.html>
- Estimating environmentally-dependent diversification rates: <https://revbayes.github.io/tutorials/divrate/env.html>
- Incorporating incomplete taxon sampling using either uniform or empirical taxon sampling: <https://revbayes.github.io/tutorials/divrate/sampling.html>

Reviewer #2 (Remarks to the Author):

Palazzesi and colleagues present a phylogenetic study of two large plant clades (daisies and grasses), which are dominant in open-vegetation habitats. They build the largest Asteraceae super-tree to date and use a published phylogeny of Poaceae to investigate diversification rates through time. The authors develop a novel statistical method to assess temporal variation in speciation and extinction rates and test whether rate changes correlate with Cenozoic changes in atmospheric CO₂. They find that changes in speciation rates in both

clades negatively correlate with changes in CO₂ (i.e. as CO₂ decreased, speciation rates increased).

The birth-death model developed in this paper models rate variation as a function of time (allowing rates to change between fixed bins in an auto-correlated fashion), and as a function of changes in CO₂ concentration (estimating correlation parameters between delta CO₂ and speciation and extinction rates). This implementation clearly improves previous models correlating diversification rates with a time-variable curve in that the null model (i.e. when correlation parameters equal zero) is a more realistic birth-death with rate shifts, rather than a constant rate process.

Query #8) While I think the manuscript is well written and the results are interesting, I also have some concerns about its design and methods. In general, I think the study should include more explicit hypothesis testing.

Response: We thank Reviewer #2 for the suggestion to use model testing. We have now included Bayes factors, as all our analyses are performed in a Bayesian statistical framework. We used state-of-the-art stepping-stone sampling and thus applied for the first time Bayesian model selection on episodic birth-death models using the GMRF and HSRMF models. We show the results in Supplementary Figure S7 and S10 (also included in this letter above). For the environmental correlation we now also compute posterior probabilities and Bayes factors, see main text Figure 3 and Supplementary Figure S10.

Query #9) The assumption of auto-correlated speciation and extinction rates is justified by the fact that “it makes sense biologically” [p. 9]. While I agree that auto-correlated rates are a plausible hypothesis, the assumption seems to be in contrast with drastic rate changes linked to e.g. mass extinctions or adaptive radiations. The existence of autocorrelation should be something assessed by hypothesis testing. Is an auto-correlated model better than the alternative birth-death model with independent rates? For instance, one could implement a “white noise” version of the episodic birth-death model, with normal priors on the log-transformed rates and hyper-priors to adapt the prior distribution to the data. This issue is analogous to auto-correlation in molecular clock models. Clock rates across branches were assumed to be auto-correlated for years, and are now mostly considered uncorrelated based on empirical evidence.

Response: We really appreciate Reviewer #2 feedback; In this new version we tested several models including two autocorrelated models and an uncorrelated model (as suggested using independent/uncorrelated lognormal distributions (UCLN) on the per-interval rates). We show the results in Supplementary Figure S9 (also included in this letter above).

Query #10) The authors test here the effect of CO₂ on diversification without testing or discussing alternative explanations. For instance, diversity dependence and other biotic interactions could have contributed or even triggered the diversification of these groups. The author mentions the effects of grassland expansion on grazer diversity, but I think reciprocal effects should be mentioned as well. Did the expansion of large herbivores (e.g. elephants) contribute to grass diversification by maintaining open habitats clear of forest or even contributing to their expansion? Temperature changes could also be a variable worth testing.

Response: We thank Reviewer #2 for the suggestion. We have now also included paleo-temperature to test for correlations to the diversification rates. Using our Bayes factor analyses, we can conclude that a correlation with CO₂ fits our observed data better than a correlation with paleo-temperature (see Figure S10 and below). We also expanded our discussion about the coevolution between large herbivores and grasslands.

Query #11) Alternative hypotheses (auto-correlation, independent-rates, CO₂, etc.) could be formally tested through Bayes factors. This would also help understanding by how much the CO₂ correlation improves the fit compared to a simpler episodic birth-death model.

Response: In this new version, we have added alternative hypotheses testing using Bayes factors as Reviewer #2 suggested. Please see Supplementary Figure S7 and S10 (also included in this letter above) and the comments above.

Query #12) The authors transform the environmental variable (CO₂ concentration) by computing the ratio between two sequential measurements and claim this approach “has

the clear advantage that our computation is insensitive to the magnitude of the environmental variable". While I agree this is a good approach for CO₂ concentration, the suitability of this approach is still dependent on the variable: for instance using ratios for temperature would not make sense and would be scale dependent (e.g. Fahrenheit vs Celsius degrees). Furthermore, using the delta-CO₂ in the analysis rather the CO₂ values, changes the interpretation of the correlation: it is not low CO₂ levels that explain increased speciation rates, rather it is the decrease of CO₂ that promotes diversification. What is the predicted speciation rate if CO₂ is low and constant?

Response: We thank Reviewer #2 for this point. We have changed our model to use the environmental variable directly as a predictor for the change in diversification rates, i.e., mapping the delta-CO₂ to the delta-lambda from the previous time interval to the next. Our new model approach also allows us to directly connect our model to the previously described model by Condamine et al (2013), see the Methods section. That is, our simplest model now is equivalent (albeit using a log-transformed notation that is more convenient to us) to the model by Condamine et al (2013).

Minor points

-Query #13) P. 8, first paragraph: I agree on discarding the second calibration scenario on the basis that the placement of microfossils is uncertain. However, the fact that the results of the second calibration scenario don't match the CO₂ dependence hypothesis (last sentence of the paragraph) should not be an argument for discarding the analysis.

Response: In our new version, we include the analysis of the second calibration scenario (#2) of grasses as a testing hypothesis and discussed the results in the main manuscript. We thank the Reviewer #2 for having noticed this.

Query #14) I suggest adding a figure to show speciation and extinction rates through time as estimated under the model that includes both time-variable rates and CO₂ dependence. This will help understand the magnitude of the effect of CO₂ on speciation rates.

Response: We have now added a new figure, see below. The left Figure below shows the diversification rates for the daisy phylogeny, and the right Figure the diversification rates for the grasses phylogeny. Importantly, we see that (a) diversification rates are driven by the data (i.e., the phylogeny) and not the specific model in the case of CO₂, and (b) diversification rate estimates disagree for paleo-temperature, which emphasizes also the conflict and uncertainty we observed in the inferred strength of correlation between diversification rates and paleo-temperature.

Query #15) What is the effect of bin size used in the episodic birth-death model on the inferred speciation and extinction rates and on the correlation factors?

Response: This is an interesting point raised by Reviewer #2. We have now included analyses using bin sizes of 1MY, 2.5MY and 5MY. The results are generally robust (see Supplementary Figure S11 and S12), although too large bin sizes deflate the correlation (see reproduced Figure S11 below).

Query #16) Figure 2 shows a cartooned example of how the auto-correlated birth-death model works showing rate shifts between some of the time bins, whereas in other bins the rate is constant. This gives the impression that the model used here actually tests whether there is rate variation or not across bins, whereas –as far as I understand– rates are assumed to be independent across time bins.

Response: Yes, we thank the reviewer to point out this confusing part in our cartoon example. We have modified the figure now and show changes between rates in each bin. We hope that this is now easier to understand.

-Query #17) Figure 3. “Daisies” is missing in the figure. Are the correlation coefficients (beta) between diversification-rates and CO₂ concentration relative to speciation (beta_lambda)?

Response: We thank the reviewer for spotting this mistake. We have added “Daisies” to the legend. We have also added more explanations and models for the correlation between diversification rates and CO₂. In these new equations, we hope that it is clear that the correlation coefficient is relative to the CO₂ but not the speciation rate.

Query #18) The authors implemented a birth-death model to incorporate non-random incomplete taxon sampling, which has quite strong effects on the estimated rates compared to the standard assumptions. The implementation is described as “similar to the approach used by Stadler and Bokma”, however it is not clear whether the method is in fact the one developed by Stadler and Bokma or if it differs from it.

Response: We apologize for the confusion. Our approach uses a different derivation but arrives at the same result as Stadler and Bokma if diversification rates are constant. Our approach for the derivation allowed us to extend this model to time-varying diversification rates, which we are primarily interested in and could not apply using the equations from Stadler and Bokma. We have changed the text accordingly.

-Query #19) It would be interesting to see in an extended figure the delta-CO₂ curve that was used to test for correlations between diversification and CO₂, i.e. the ratios between CO₂ at adjacent bins.

Response: We added a figure as suggested. Please refer to Figure S3.

-Query #20) Adding equation numbers, page numbers, line numbers would help future revisions.

Response: We apologize for this omission. We now have added equation numbers, page numbers, line numbers.

Reviewer #3 (Remarks to the Author):

This study presents a Bayesian phylogenetic approach that uses time-calibrated phylogenies to simultaneously estimate diversification rates through time and correlations between an environmental variable and lineage diversification. This method is then used to examine the influence of past CO₂ concentrations on the diversification of Poaceae and Asteraceae, two highly diverse grassland clades. They find accelerating diversification in both clades from ~28-15Ma, a short period of lower diversification rates and then further increases beginning ~10Ma.

This is a well-written paper and I like the overall approach that compares the diversification of Asteraceae and Poaceae to one another. The described method is powerful and will be useful for a wide variety of biological questions. I also appreciate the nuanced way that this paper discusses diversification—the authors never state that the drop in CO₂ concentrations drives diversification, only that they are correlated.

Query #21) This analysis differs from previous studies of diversification in Poaceae in that it looks at shifts in diversification through time rather than among lineages. I would have liked to see a supplementary figure with diversification rates plotted on each phylogeny (Poaceae and Asteraceae)—maybe just simplified trees that have the major clades? It would then be possible to compare the timing of the shifts observed more directly to previous results. I also think that there should be some discussion about the simplifications that this model makes—specifically that all members of the clade are diversifying at the same rate at a given time. Both Asteraceae and Poaceae are diverse, cosmopolitan clades that contain a lot of physiological variation—a single diversification rate is probably summing over a lot of variation among lineages.

Response: The plot that Reviewer #3 suggests can be conducted by other models within RevBayes, for example the Branch-Specific Diversification Rate Estimation method, using a sampled phylogeny. However, our main goal here was to integrate missing species—which accounts for about ~70% in Poaceae and ~90% in Asteraceae—in order to estimate the most important shifts in diversification rates through time and link them to environmental factors. At present, unfortunately, there are no existing methods (to our knowledge) to plot changes in diversification rates among lineages with such a large number of missing species. We assume that estimating diversification rates of these speciose families using sampled phylogenies (with the majority of the species missing) will underestimate and/or misplace the diversification rates among lineages. While we agree that both Asteraceae and Poaceae

contain a lot of physiological variation, these families are particularly well represented in open-habitat ecosystems. In this new version, however, we included a new Supplementary Figure 4 plotting simultaneously chronograms of grasses and daisies along with their net-diversification rates, speciation rates and extinction rates through time using our empirical taxon sampling.

-Query #22) I appreciate that the authors have chosen not to focus on C4 photosynthesis, instead taking a wider view of grassland clades more generally; however, not including at least some discussion of how C4 photosynthesis is related to the observed patterns seems strange to me. C4 photosynthesis is generally believed to be an essential trait that has contributed to the global dominance of Poaceae and is highly related to plant performance under low atmospheric CO₂.

Response: We have now restructured the discussion and included the evolution of grasslands in the context of the rise of the C₄ photosynthetic pathway.

I had a few more specific comments as well:

Query #23) “We selected the chronogram calibrated using the megafossil (calibrated scenario #1) because the phylogenetic placement of phytoliths is somewhat controversial¹⁶. . .”

The better reference for the implications/controversy of the phytolith Poaceae dates is Christin et al. 2014 (# 43). The choice here to use the macro fossil dating scenario seems reasonable and matches the choices made for Asteraceae, but I think the justification for this choice should be described better. It’s an important decision because the phytolith dating scenario pushes back the ages for most of the Poaceae tree and it could significantly affect the results that are presented. With the older dates would the increase in Poaceae diversification predate the drop in CO₂? It seems from Figure 3 that it might not, but I think it would strengthen the argument of the paper if the results were robust to these dates. It’s true that the phylogenetic placement of phytoliths is controversial because there is still uncertainty about the extent of homoplasy/their phylogenetic distribution, but they do provide useful information in clade with very little macro fossil information and discounting them entirely makes me uncomfortable.

Response: We have now included the Christin et al. 2014 reference and analyzed the chronogram with the #2nd calibration scenario. As Reviewer #3 properly emphasizes, the major diversification shift using the older calibration scenario postdates the most important drop in CO₂. We appreciate Reviewer #3 for flagging this up.

-Query #24) “Furthermore, the origin of C₄ lineages within Poaceae under the scenario #2 occurred well before the drop of CO₂. . .”

The only C_4 origin that is well before the drop of CO_2 under dating scenario #2 is in the Chloridoideae. All other C_4 origins occur near or after the CO_2 drop.

Response: We have now modified the text.

Query #25) Figure 3. The yellow line (Asteraceae) is not labeled in the figure legend.

Response: We have now labeled Asteraceae.

Query #26) “The available experimental evidence from plants grown under low atmospheric CO_2 shows that plant performance deteriorates rapidly as CO_2 declines. . .”

More accurately low CO_2 limits performance of some taxa—responses vary significantly among species.

Response: We modified the text following Reviewer #3 suggestions.

“The low CO_2 conditions probably imposed a selective pressure for life history traits; the main plant groups of the grassy biome—mostly annual or biannual herbs with short generation times. . .became extinct.”

This section is confusing to me. Ecological models and experimental data suggest that high CO_2 favors woody plants over grasses (e.g. Kgope et al. 2009 ‘Growth responses of African savannah trees implicate atmospheric CO_2 as a driver of past and current changes in savanna tree cover; Bond & Midgley 2000 ‘A proposed CO_2 –controlled mechanism of woody plant invasion in grasslands and savannas; Bond & Midgley 2011 ‘Carbon dioxide and the uneasy interactions of trees and savannah grasses’), but this is separate from arguments related to generation time—Poaceae and Asteraceae species have many physiological adaptations to living in open environments. It makes sense, however, that short generation times might have enabled rapid evolution which could have facilitated their adaptation into new environments. Maybe this section would be clearer if ‘open systems’ or some alternative was used rather than ‘grassy biomes.’ It’s hard to talk about grasses evolving to occupy biomes that are already characterized by their presence.

It would be useful if the ways that this method differs from previous methods (refs 22-24) were more explicitly stated. As it is, it's somewhat hard to tell.

Response: We have now modified the text and clarified this discussion. We also changed the term “grassy biomes” as Reviewer #3 suggested; we again thank the reviewer for her/his suggestions; all of them helped improve our manuscript.

References

- Christin, P.-A. et al. 2014. Molecular Dating, Evolutionary Rates, and the Age of the Grasses. *Syst. Biol.* 63, 153–165.
- Condamine, F. L., Rolland, J. & Morlon, H. 2013. Macroevolutionary perspectives to environmental change. *Ecol. Lett.* 16, 375 72–85.
- Höhna, S., 2014. Likelihood inference of non-constant diversification rates with incomplete taxon sampling. *PLoS one*, 9(1), p.e84184.
- Höhna, S., Stadler, T., Ronquist, F. and Britton, T., 2011. Inferring speciation and extinction rates under different sampling schemes. *Molecular biology and evolution*, 28(9), pp.2577-2589.
- Stadler, T. & Bokma, F. 2013. Estimating speciation and extinction rates for phylogenies of higher taxa. *Syst. Biol.* 62, 220–230 426.

Reviewers' Comments:

Reviewer #1:

Remarks to the Author:

I was Reviewer 1 of the previous version of this manuscript. I appreciate that the authors have put in a lot of work to address my various earlier comments. The additional simulation tests are certainly valuable, and the RevBayes tutorial is even more useful than the typical procedure of just archiving code for a particular analysis. The paper will overall be a very nice contribution.

My one new concern is that the identifiability issues raised by Louca & Pennell (doi:10.1038/s41586-020-2176-1) should be addressed. The present analysis definitely seems to be type they are concerned about, with complex time-varying diversification rates inferred from divergence times/LTT plots. But perhaps the authors can explain if the problem is averted by looking at correlations with CO2 and temperature over time, which provides a priori hypotheses that are tested, rather than fitting a model with arbitrary temporal variation in diversification.

For the correlation parameter beta, my first reaction looking at Fig 3b top panels is that the effect size is extremely small. Normally I would expect a "correlation coefficient" to range between -1 and 1. But the Methods section later explains what beta really means. It would help if the authors could provide some context for interpreting the inferred strength of correlation. For example, how much does CO2 explain relative to what remains unexplained?

Fig 1 caption: "specious" is definitely not the correct word here

Reviewer #2:

Remarks to the Author:

The authors did an excellent job revising their paper, which I find very interesting and well presented. I only have a couple of remaining comments, which I think should be addressed in a further minor revision.

The first one relates to the findings of Louca and Pennell (2020 Nature), which I am sure the authors are aware of and which demonstrate cases of unidentifiability in rate variation through time. While I do not think the issues they raise invalidate this study, this problem should still be touched upon in the paper. For example: is it possible that the different diversification patterns inferred through the UCLN model (line 50) is due to unidentifiability sensu Louca&Pennell?

Line 60: the same argument made here could be used to say that the first analysis (excluding the Cretaceous calibration) should be taken with caution because it excludes a potentially key fossil record. Can you provide more information about scenario #1 should be preferred? Otherwise you can just present them as alternative scenarios without favoring one.

Line 126: "next 50 years" [s missing]

Line 137: can you clarify what a "spatial issue" is?

Line 327: Please make the scripts/commands used to set up the RevBayes analyses available as supplementary data along with a specification of which version of the program was used here. This is essential to ensure the reproducibility of the analyses. Also: following this link <https://revbayes.github.io/tutorials.html> I got a 404 error.

REVIEWER COMMENTS

Reviewer #1 (Remarks to the Author):

I was Reviewer 1 of the previous version of this manuscript. I appreciate that the authors have put in a lot of work to address my various earlier comments. The additional simulation tests are certainly valuable, and the RevBayes tutorial is even more useful than the typical procedure of just archiving code for a particular analysis. The paper will overall be a very nice contribution.

Response: We really appreciate Reviewer #1 feedback. Please, thank her/him on our behalf.

My one new concern is that the identifiability issues raised by Louca & Pennell (doi:10.1038/s41586-020-2176-1) should be addressed. The present analysis definitely seems to be the type they are concerned about, with complex time-varying diversification rates inferred from divergence times/LTT plots. But perhaps the authors can explain if the problem is averted by looking at correlations with CO₂ and temperature over time, which provides a priori hypotheses that are tested, rather than fitting a model with arbitrary temporal variation in diversification.

Response: Louca & Pennell (2020) have shown that, in general, extant timetrees alone cannot be used to reliably infer speciation rates, extinction rates or net diversification rates if arbitrarily complex speciation and extinction rate functions are used. As the reviewer points out, testing specific competing hypotheses is nevertheless possible, as recently published by Helmstetter et al. (2021). Furthermore, the episodic birth-death process (or also called piecewise-constant birth-death process) is actually identifiable, compared with arbitrary continuous rate functions, as shown by Legried and Terhorst (2021). We have added the following sentences to the discussion:

“Recently, Louca and Pennell²⁸ showed that phylogenies of extant taxa are consistent with infinitely many diversification rate models and therefore diversification rates are not identifiable if arbitrarily complex diversification rate functions are allowed. Our diversification models, on the other hand, are identifiable because of the piecewise-constant (episodic) diversification rates model²⁹. Furthermore, model comparison is robust when well-formulated alternative hypotheses are used³⁰, as is the case for the comparison between different environmentally-dependent diversification models³¹.” (lines 52-57)

For the correlation parameter beta, my first reaction looking at Fig 3b top panels is that the effect size is extremely small. Normally I would expect a "correlation coefficient" to range between -1 and 1. But the Methods section later explains what beta really means. It would help if the authors could provide some context for interpreting the inferred strength of correlation. For example, how much does CO₂ explain relative to what remains unexplained?

Response: We are sorry about the confusion. We have replaced “correlation coefficient” with “correlation factor” to hopefully avoid confusion. We have also added a sentence in the caption of Figure 3 to explain the magnitude of the correlation factor. In diversification models (also state-dependent diversification rates), one cannot say how much the variation in diversification rates is explained by the predictor variable. Unfortunately, there is no such information available.

“Note that the correlation factor β does not follow the same scale as common correlation coefficients (which are between -1 and 1) but instead represents the factor by which to transform the environmental variable into diversification rates (see methods).” (in Fig 3 caption).

Fig 1 caption: "specious" is definitely not the correct word here

Response: We have now modified the word by “species-rich”.

Reviewer #2 (Remarks to the Author):

The authors did an excellent job revising their paper, which I find very interesting and well presented. I only have a couple of remaining comments, which I think should be addressed in a further minor revision.

Response: We really appreciate Reviewer #2 feedback. Again, please, thank her/him on our behalf.

The first one relates to the findings of Louca and Pennell (2020 Nature), which I am sure the authors are aware of and which demonstrate cases of unidentifiability in rate variation through time. While I do not think the issues they raise invalidate this study, this problem should still be touched upon in the paper. For example: is it possible that the different diversification patterns inferred through the UCLN model (line 50) is due to unidentifiability sensu Louca&Pennell?

Response: We thank the reviewer for raising this concern which most likely many readers will have too. We have addressed the non-identifiability issue raised by Louca and Pennell (see comment to Reviewer 1). In short, the different pattern observed from the UCLN model is not due to the non-identifiability issue because piecewise-constant rates, as used here, are identifiable (Legried and Terhorst (2021)). If the pattern observed from the UCLN model would stem from non-identifiability of the likelihood function, then our posterior estimates would be driven by the prior distribution. This would lead to estimates clustered around the prior mean which is identical for all epochs, but we clearly did not observe such a behaviour.

Line 60: the same argument made here could be used to say that the first analysis (excluding the Cretaceous calibration) should be taken with caution because it excludes a potentially key fossil record. Can you provide more information about scenario #1 should be preferred? Otherwise you can just present them as alternative scenarios without favoring one.

Response: We have now presented more explicitly both alternative scenarios without favoring one (lines 64-70) and shown testing results.

Line 126: “next 50 years” [s missing]

Response: Modified.

Line 137: can you clarify what a “spatial issue” is?

Response: We have now exemplified some geospatial issues (line 154).

Line 327: Please make the scripts/commands used to set up the RevBayes analyses available as supplementary data along with a specification of which version of the program was used here. This is essential to ensure the reproducibility of the analyses. Also: following this link <https://revbayes.github.io/tutorials.html> I got a 404 error.

Response: We are sorry about the typo in the general link to the tutorial. The correct link is <https://revbayes.github.io/tutorials/>. All scripts to set up the RevBayes analysis are available in the following links <https://revbayes.github.io/tutorials/divrate/env.html> and <https://revbayes.github.io/tutorials/divrate/sampling.html>. These are now included in our main manuscript along with the version of RevBayes used to conduct our analyses.

Reviewer #1 was also asked to check the responses to Reviewer #3, who was unable to comment. These are Reviewer #1's further comments regarding your numbered responses to Reviewer #3:

#21

I think the authors slightly misunderstood the requested figure. It would be like the new Fig S4, with the same rates over time, but with a greatly-simplified tree that depicts triangles sized proportionately to the number of species (including those not on the tree) in various clades. This change to the tree would also largely alleviate the authors' concern that the diversification rates are based on all species, not only those on the tree.

Response: Indeed we were not sure if we understood the suggested figure correctly. To be sure to address the suggestion we have now included two new figures to the Supplement; Fig S4 and Fig S5. One of them (Fig S4) depicts the rate shifts among lineages (that is, coloured transition in a phylogeny) and the other (Fig S5) the rate shifts against simplified trees for both families, with triangles indicating the most important lineages, as Reviewer

suggested. The new addition of these plots shows more clearly the timing of the diversification of the most species-rich clades of daisies and grasses.

#21.a The other part of this comment was that a mention be added about the model assuming the same rate for the whole family at each time. This seems like a very reasonable request that was not met.

Response: We are sorry that we overlooked this request. We have now added the following sentence to the introduction to make as early as possible clear what the model assumption is:

“This model assumes that diversification rates are homogeneous (equal for all lineages at the same time) and does not allow for lineage-specific shifts in diversification rates.” (line 39-40)

This model assumption is also highlighted again in the methods section:

“We model diversification rates (i.e., speciation and extinction rates) as constant within an interval but independent between intervals, where intervals are demarcated by instantaneous rate-shift events, and equal among contemporaneous lineages.” (lines 195-197)

#22

This discussion has been added and seems reasonable, but I'm not informed enough to comment on whether exactly the right points were mentioned.

#23-27

These points seem to have been addressed sufficiently.

Response: We thank Reviewer #1 to have revised Reviewer's #3 main concerns. Please, thank her/him on our behalf.

References

Helmstetter, A.J. et al. Pulled Diversification Rates, Lineages-Through-Time Plots and Modern Macroevolutionary Modelling, *Systematic Biology*, 2021; syab083, <https://doi.org/10.1093/sysbio/syab083>.

Legried, B. and Terhorst, J., 2021. A class of identifiable birth-death models. *bioRxiv*.

Louca, S., Pennell, M.W. Extant timetrees are consistent with a myriad of diversification histories. *Nature* 580, 502–505 (2020). <https://doi.org/10.1038/s41586-020-2176-1>.

Reviewers' Comments:

Reviewer #1:

Remarks to the Author:

I was Reviewer 1 before. The authors have adequately addressed my previous comments. In particular, Fig S4 is very nice, and the explanation about identifiability seems fine. I'm a bit disappointed there's no better intuitive explanation of the the magnitude of beta, but I can't think of one either. Overall, a nice paper!

RESPONSE LETTER:

Reviewer #1 (Remarks to the Author):

I was Reviewer 1 before. The authors have adequately addressed my previous comments. In particular, Fig S4 is very nice, and the explanation about identifiability seems fine. I'm a bit disappointed there's no better intuitive explanation of the the magnitude of beta, but I can't think of one either. Overall, a nice paper!

Response: We really appreciate Reviewer #1 comments on our article. Please, thank her/him on our behalf. We do think our manuscript has been improved by her/his feedback.